

# Global simulation of tropospheric chemistry at 12.5 km resolution: performance and evaluation of the GEOS-Chem chemical module (v10-1) within the NASA GEOS Earth System Model (GEOS-5 ESM)

Lu Hu[1,2], Christoph A. Keller[2,3,4], Michael S. Long[2], Tomás Sherwen[5,6], Benjamin Auer[3,7], Arlindo Da Silva[3], Jon E. Nielsen[3,7], Steven Pawson[3], Matthew A. Thompson[3,7], Atanas L. Trayanov[3,7], Katherine R. Travis[2,8], Stuart K. Grange[6], Mat J. Evans[5,6], and Daniel. J. Jacob[2]

[1]Department of Chemistry and Biochemistry, Missoula, MT, USA
[2]John A. Paulson School of Engineering and Applied Science, Harvard University, Cambridge, MA, USA
[3]Global Modeling and Assimilation Office, NASA Goddard Space Flight Center, Greenbelt, MD, USA
[4]Universities Space Research Association, Columbia, MD, USA
[5]National Centre for Atmospheric Science, University of York, York, UK
[6]Wolfson Atmospheric Chemistry Laboratory, University of York, York, UK
[7]Science Systems and Applications, Inc., Lanham, MD, USA
[8]Department of Civil and Environmental Engineering, Massachusetts Institute of Technology, Cambridge, MA, USA

**Correspondence:** Lu Hu (lu.hu@mso.umt.edu)

**Abstract.** We present a full-year on-line global simulation of tropospheric chemistry (158 coupled species) at cubed-sphere c720 ($\sim 12.5 \times 12.5 \, \text{km}^2$) resolution in the NASA GEOS Earth System Model, Version 5 (GEOS-5 ESM) with GEOS-Chem as a chemical module (G5NR-chem). The GEOS-Chem module within GEOS uses the exact same code as the off-line GEOS-Chem chemical transport model (CTM) developed by a large atmospheric chemistry research community. In this way, continual

5 updates to the GEOS-Chem CTM by that community can be seamlessly passed on to the GEOS chemical module, which remains state-of-the-science and referenceable to the latest version of GEOS-Chem. The 1-year G5NR-chem simulation was conducted to serve as Nature Run for observing system simulation experiments (OSSEs) in support of the future geostationary satellite constellation for tropospheric chemistry. It required 31 walltime days on 4707 compute cores with only 24% of the time spent on the GEOS-Chem chemical module. Results from the GEOS-5 Nature Run with GEOS-Chem chemistry were shown to

10 be consistent to the off-line GEOS-Chem CTM and were further compared to global and regional observations. The simulation shows no significant global bias for tropospheric ozone relative to the Ozone Monitoring Instrument (OMI) satellite, and is highly correlated with observations spatially and seasonally. It successfully captures the ozone vertical distributions measured by ozonesondes over different regions of the world, as well as observations for ozone and its precursors from the Aug-Sep 2013 SEAC[4]RS aircraft campaign over the Southeast US. It systematically overestimates surface ozone concentrations by $10 \, \text{ppbv}$

15 at sites in the US and Europe, a problem currently being addressed by the GEOS-Chem CTM community and from which the GEOS ESM will benefit through the seamless update of the on-line code.





# 1 Introduction

Integration of atmospheric chemistry into Earth Science Models (ESMs) has been identified as a next frontier for ESM development (National Research Council, 2012) and is as a priority science area for atmospheric chemistry research (National Academies of Sciences, Engineering, and Medicine, 2016). Atmospheric chemistry drives climate forcing and feedbacks, is

an essential component of global biogeochemical cycling, and is key to air quality applications. A growing ensemble of atmospheric chemistry observations from space needs to be integrated into ESM-based data assimilation systems. Models of atmospheric chemistry are rapidly evolving, and an atmospheric chemistry module within an ESM must be able to readily update to the state-of-the-science. We have developed such a capability by integrating the GEOS-Chem chemical transport model (CTM) as a comprehensive and seamlessly updatable atmospheric chemistry module in the NASA Goddard Earth Ob-

serving System (GEOS) ESM (Long et al., 2015). Here we present the first application and evaluation of this GEOS-Chem capability within GEOS Version 5 (GEOS-5) for a full-year global simulation of tropospheric ozone chemistry at cubed-sphere c720 (~$12.5 \times 12.5 \, \text{km}^2$) resolution. This simulation is now serving as Nature Run (pseudo-atmosphere) for observing system simulation experiments (OSSEs) in support of the near-future geostationary satellite constellation for tropospheric chemistry (Zoogman et al., 2017).

GEOS-Chem (http://geos-chem.org) is an open-source global 3-D Eulerian model of atmospheric chemistry driven by GEOS-5 assimilated meteorological data. It includes state-of-science capabilities for tropospheric and stratospheric gas-aerosol chemistry (Eastham et al., 2014; Hu et al., 2017), with additional capabilities for aerosol microphysics (Trivitayanurak et al., 2008; Yu and Luo, 2009). It is used by over a hundred active research groups in 25 countries around the world for a wide range of atmospheric chemistry applications, providing a continual stream of innovation (Hu et al., 2017). Strong version control and

benchmarking maintain the integrity and referenceability of the model. The code is freely available through an open license (http://acmg.seas.harvard.edu/geos/geos_licensing.html).

GEOS-Chem as used by the atmospheric chemistry community operates in an "off-line" CTM mode, without explicit simulation of meteorology. Meteorological data are input to the model to simulate chemical transport and other processes. The off-line approach makes the model simple to use, and facilitates community development of the core chemical module that describes

local chemical sources and sinks from emissions, reactions, thermodynamics, and deposition. Implementing the GEOS-Chem chemical module into ESMs offers a state-of-the-science and referenceable representation of atmospheric chemistry, but it is essential that the module be able to automatically incorporate new updates as the off-line model evolves. Otherwise it would become quickly dated and unsupported.

From a GEOS-Chem atmospheric chemistry user perspective, there are a number of reasons why an on-line simulation

capability is of interest. Users working on both climate and atmospheric chemistry modeling can use the same module for both, improving the consistency of their approach. Some atmospheric chemistry problems involving fast coupling between chemistry and dynamics, such as aerosol-cloud interactions, require on-line coupling. As the resolution of ESMs increases, use of archived meteorological data becomes more difficult and incurs increased error (Yu et al., 2018), so on-line simulation can become more attractive. Finally, on-line simulations can take advantage of vast computational resources available at climate





modeling centers to achieve very high resolution as illustrated in this paper. Such high-resolution model outputs are particularly important for air quality and human health applications (Cohen et al., 2017) and for OSSEs to support the design of satellite missions (Claeyman et al., 2011; Fishman et al., 2012; Yumimoto, 2013; Zoogman et al., 2014; Barré et al., 2016). A future *modus operandi* for the GEOS-Chem community might involve development in the off-line model at coarse resolution, and

on-line simulations when high resolution is required.

We have developed the capability to efficiently integrate the GEOS-Chem chemical module into ESMs in a way that enables seamless updating to the latest standard version of GEOS-Chem. This involved transformation of GEOS-Chem into a grid-independent and Earth System Modeling Framework (ESMF)-compliant model (Long et al., 2015). The exact same code is now used in stand-alone off-line mode (CTM) by the GEOS-Chem community and as an on-line chemical module within

GEOS-5 (Figure 1). Using the exact same code for both applications ensures that the chemical module in the ESM keeps current with the latest well-documented standard version of GEOS-Chem. An important component is the Harvard-NASA EMission COmponent (HEMCO; Keller et al., 2014), which allows GEOS-Chem users to build customized layers of emission inventories on any grid and with no editing of the GEOS-Chem source code. HEMCO is now the standard emission module in GEOS-Chem and is used in GEOS-5 as an independent module for applications when full chemistry is not needed.

Here we apply the GEOS-Chem chemical module within GEOS-5 in a very-high-resolution (VHR) simulation of global tropospheric ozone chemistry. The simulation is conducted for one full year (2013-2014) at c720 cubed-sphere resolution ($\sim 12.5 \times 12.5 \, \mathrm{km}^2$) with 158 coupled chemical species. To the best of our knowledge, such a resolution in a state-of-the-science global simulation of tropospheric chemistry is unprecedented. Resolution not only increases the quality of local information, e.g., for air quality, but it also provides better representation of chemical non-linearities. We compare the model outputs with

coarse off-line GEOS-Chem CTM results and with independent observations for tropospheric ozone and precursors as a test of fidelity and increased power. We refer to this VHR simulation as GEOS-5 Nature Run with GEOS-Chem chemistry, or G5NR-chem..

## 2   GEOS-Chem as an atmospheric chemistry module in the GEOS ESM

An Eulerian (fixed frame of reference) CTM such as GEOS-Chem solves the system of coupled continuity equations for an

ensemble of $m$ species with number density vector $\mathbf{n} = (n_1, \ldots, n_m)^T$,

$$\frac{\partial n_i}{\partial t} = -\nabla \cdot (n_i \mathbf{U}) + (P_i - L_i)(\mathbf{n}) + E_i - D_i \qquad i \in [1, m] \tag{1}$$

where $\mathbf{U}$ is the wind vector including subgrid components to be parameterized as turbulent diffusion or convection. $(P_i - L_i)(\mathbf{n})$, $E_i$, and $D_i$ are the local chemical production and loss, emission, and deposition rates respectively of species $i$. Coupling across species is through the chemical term $(P_i - L_i)$. In GEOS-Chem, as in all 3-D CTMs, Eq. 1 is solved by operator

splitting to separate the transport and local components over finite time steps. The local operator,

$$\frac{dn_i}{dt} = (P_i - L_i)(\mathbf{n}) + E_i - D_i \qquad i \in [1, m] \tag{2}$$



includes no transport terms (no spatial coupling) and thus reduces to a system of coupled ordinary differential equations (ODEs). It is commonly called the chemical operator even though emission and deposition terms are included. The transport operator,

$$\frac{\partial n_i}{\partial t} = -\nabla \cdot (n_i \mathbf{U}) \qquad i \in [1, m] \tag{3}$$

does not involve coupling between chemical species.

Use of GEOS-Chem as chemical module in an ESM requires only the code that updates concentrations over a given time step for local production and loss as given by Eq. 2 (Figure 1). The CTM has its own transport modules to solve Eq. 3 using archived meteorological inputs (off-line), but these are not needed in the ESM where transport is computed as part of atmospheric dynamics (on-line).

The ESM chemical module is tasked with updating chemical concentrations by integration of Eq. 2 on the ESM grid and timestep. Exchange of information between the chemical module and other ESM modules can be done by various couplers such as ESMF (Hill et al., 2004). Our guiding principle is that the CTM and the ESM chemical module share the exact same code. This required restructuring GEOS-Chem to a grid-independent form, and making the code compliant with the ESMF Modeling Analysis and Prediction Layer (MAPL) coupler used by GEOS (Long et al., 2015). 1-D vertical columns are the

smallest efficient unit of computation for the chemical module because several operations are vertically coupled including radiative transfer, vertically distributed emissions, wet scavenging, and particle settling. In the now grid-independent GEOS-Chem code, horizontal grid points are selected at runtime through the ESMF interface. The chemical and emission modules proceed to solve Eq. 2 on 1-D columns for the specified horizontal gridpoints (Figure 1). We managed to carry out this major software transformation in GEOS-Chem in a way that was completely transparent to CTM users (Long et al., 2015). The exact

same ESMF-compliant, grid-independent GEOS-Chem code is now used both in the stand-alone CTM and within GEOS. This enables seamless integration of future new GEOS-Chem scientific developments into the GEOS chemical module, which thus always remains current and referenced to the latest standard version of GEOS-Chem.

An important step in transforming GEOS-Chem to a grid-independent structure was to reconfigure the emission module. The emission module consists of multiple layers of databases and algorithms describing emissions for different species and regions,

with scaling factors defining diurnal/weekly/seasonal/secular trends or dependences on environmental variables. The databases are on different grids and time stamps, and may add to or supersede each other as controlled by the user. The HEMCO module allows users to choose any combination of emission inventories in NetCDF format, on any grid and with any scaling factors, and apply them to any model grid specified at runtime (Keller et al., 2014). HEMCO provides a complete functional separation of emissions from transport, deposition, and chemistry in GEOS-Chem. The purpose of this separation is that the ESM may need

emissions independently from atmospheric chemistry, for example to simulate species such as $CO_2$ and methane. HEMCO is thus configured as a stand-alone component in the ESM, accessed separately through the ESMF interface (Figure 1).

Some care is needed in when interfacing the GEOS-Chem chemical module with fast vertical transport processes in GEOS-5 involving boundary layer mixing and deep convection with wet scavenging. Here we apply boundary layer mixing at every time step in GEOS-5 with emissions and dry deposition updates from GEOS-Chem, but before applying chemistry. This avoids





anomalies in the lowest model layer when the timescale for boundary layer mixing is shorter than the timestep for emissions. Deep convective transport of chemical species including scavenging in the updrafts is performed by the GEOS-Chem convection scheme driven by instantaneous diagnostic variables from the GEOS-5 convection component (Molod et al., 2015). This takes advantage of the gas and aerosol scavenging capability of the GEOS-Chem scheme (Liu et al., 2001; Amos et al., 2012). Radon-222 tracer simulation tests within GEOS-5 show that the GEOS-Chem convection scheme closely reproduces the GEOS-5 convective transport (Yu et al., 2018).

## 3 GEOS-5 Nature Run with GEOS-Chem chemistry

We perform a year-long (July 1 2013 to July 1 2014) GEOS-5 simulation with GEOS-Chem at cubed-sphere c720 (~$12.5 \times 12.5 \, \mathrm{km}^2$) horizontal resolution and 72 vertical levels extending up to $0.01 \, \mathrm{hPa}$. For initialization, we use 12 months at c48 resolution (~$200 \times 200 \, \mathrm{km}^2$) followed by 6 months at c720 resolution. Figure 2 shows a snapshot of the simulated $500 \, \mathrm{hPa}$ ozone field, illustrating the fine detail enabled by the very high resolution.

### 3.1 General description

The GEOS-5 Nature Run with GEOS-Chem chemistry is performed with the Heracles version of GEOS-5 (tag 'M2R12K-3_0_GCC'). The finite-volume dynamics is run in a non-hydrostatic mode with a heartbeat time of 300 seconds applied to the physics and dynamics components. The simulation is forced by downscaled meteorological data from the lower resolution MERRA-2 reanalysis (Molod et al., 2015; Gelaro et al., 2017). The downscaling is performed by using the replay capability in GEOS, which adds a forcing term to the model equations, constraining them to a specific trajectory to simulate the 2013-2014 meteorological year (Orbe et al., 2017). Downscaling applications filter the replay increments so that only the larger scales of the flow are constrained, allowing scales finer than the analysis to evolve freely. In this simulation, a wave number of 60 was chosen as the cutoff. The simulation is performed in two segments, the first with GEOS-Chem turned off ('regular replay'). The analysis increment produced during the first run segment is saved and reused in the subsequent run with GEOS-Chem turned on ('exact increment'). This two segment process is computationally more efficient as it avoids rewinding and checkpointing the model with full chemistry during the regular replay stage.

The chemical module is GEOS-Chem version v10-01 in tropospheric-only mode. It includes detailed $HO_x$-$NO_x$-VOC-ozone-$BrO_x$-aerosol tropospheric chemistry with 158 species and 412 reactions, following JPL and IUPAC recommendations for chemical kinetics (Sander et al., 2011) and updates for BrOx and isoprene chemistry (Parrella et al., 2012; Mao et al., 2013). The Fast-JX scheme is used to calculate photolysis frequencies (Bian and Prather, 2002) as implemented by (Mao et al., 2010). Linearized stratospheric chemistry is used (McLinden et al., 2000; Murray et al., 2012).The dry deposition calculation is based on a resistance-in-series model (Wesely, 1989) as implemented by Wang et al. (1998). Wet scavenging of aerosols and gases is as described by Liu et al. (2001) and Amos et al. (2012).

Emissions are calculated through HEMCO v1.1.008 (Keller et al., 2014). They are the default 2013-2014 emissions for GEOS-Chem (see Table 1 in Hu et al., 2017), with a few modifications. Open fire emissions are from the Quick Fire Emissions





Dataset (QFED) version 2.4-r6 (Darmenov and Da Silva, 2015). US $NO_x$ emissions follow Travis et al. (2016). Parameterizations for lightning $NO_x$ (Murray et al., 2012) and mineral dust aerosol emissions (Zender et al., 2003) have large dependencies on grid resolution, and are scaled globally following general GEOS-Chem practice to annual totals of $6.5\,TgN$ for lightning and $840\,TgC$ for dust. No adjustments are made to emission of biogenic VOCs (MEGANv2.1; Guenther et al., 2012; Hu et al.,

2015b) and sea salt aerosol (Jaeglé et al., 2011), both of which agree with GEOS-Chem emissions within 15%. GEOS-Chem uses in-plume chemistry of ship emissions (PARANOx) to account for the excessive dilution of ship exhaust plumes at coarse model resolution (Vinken et al., 2011); this was disabled in the VHR simulation given that the Nature Run resolution is fine enough to resolve the non-linear chemistry associated with ship plume emissions. A summary of the various emission sources used for the simulation is given in Table A2.

## 3.2 Computational environment and cost

The computations were carried out on the Discover supercomputing cluster of the NASA Center for Climate Simulation (https://www.nccs.nasa.gov/services/discover). One day of simulation took approximately 2 walltime hours, using 4705 compute cores with 45% spent on dynamics and physics, 24% on chemistry, and 31% on input/output (I/O) (Table A1). The large I/O wall time is due to bottlenecks in the MAPL software version used for the simulation, with excessive reading and remapping

of the hourly high-resolution emission fields. This issue has been addressed in newer versions of MAPL. As first shown by Long et al. (2015), the chemical module has excellent scalability even when running with thousands of cores. The percentage of the walltime spent on chemistry in the G5NR-chem (24%) is much lower than in coarse resolution simulations that are typically done with only a small number of cores (Eastham et al., 2018). The computational cost of chemistry relative to dynamics/transport decreases as grid resolution increases, thus it is no longer the computing bottleneck in ESM simulations.

# 4 Model evaluation

## 4.1 Observational datasets and off-line CTM

The GEOS-5 Nature Run with GEOS-Chem chemistry simulation is intended to support a geostationary constellation OSSEs focused on tropospheric ozone and related satellite measurements (Zoogman et al., 2017), and ozone is therefore our evaluation focus. We use 2013-2014 observations that were previously compared to the GEOS-Chem CTM including (1) global

ozonesondes and OMI (Ozone Monitoring Instrument) satellite data (Hu et al., 2017), (2) aircraft data for ozone and precursors from the NASA SEAC[4]RS campaign over the Southeast US (Travis et al., 2016), and (3) surface ozone monitoring data over Europe and the US (Yan et al., 2016; Grange, 2017). An important goal of the evaluation here is to examine consistency between the GEOS-Chem chemical module within the GEOS ESM c720 environment and the off-line GEOS-Chem CTM. Although the GEOS-Chem simulation is at coarser resolution and off-line transport may incur errors (Yu et al., 2018), it is

extensively diagnosed by the GEOS-Chem user community including recently by Hu et al. (2017) for global tropospheric ozone. Two GEOS-Chem CTM v10-01 simulations are used for comparison to the G5NR-chem: a global simulation with



$2° \times 2.5°$ resolution, and a nested simulation for North America with $0.25° \times 0.3125°$ resolution. Both are driven by GEOS-5 Forward-Processing (GEOS-5.7.2 and later versions) assimilated meteorological data. Some differences with the G5NR-chem are to be expected because of differences in the transport modules, resolution, distribution of natural sources computed on-line such as lightning $NO_x$, and meteorological data from different versions of the GEOS system (MERRA-2 vs GEOS-FP). All
comparisons to observations use model output sampled at the location and time of observations.

Observational datasets are described in the above references. Briefly, global ozonesonde observations are extracted from the WOUDC (World Ozone and Ultraviolet Data Center, http://www.woudc.org) and the NOAA ESRL-GMD (Earth System Research Laboratory – Global Monitoring Division, ftp://ftp.cmdl.noaa.gov/ozwv/Ozonesonde/). Ozonesonde stations are grouped into coherent regions for model evaluation (Tilmes et al., 2012). OMI middle tropospheric ozone data at $700 - 400\,hPa$
are from the Smithsonian Astrophysical Observatory (SAO TROPOZ) retrieval (Liu et al., 2010; Huang et al., 2017) and are regridded to $2° \times 2.5°$ resolution to reduce retrieval noise. The NASA SEAC[4]RS data set for Southeast US described by Toon et al. (2016) are filtered following Travis et al. (2016) to remove open fire plumes ($CH_3CN > 200\,pptv$), stratospheric air ($O_3/CO > 1.25\,mol\,mol^{-1}$), and urban plumes ($NO_2 > 4\,ppbv$). Hourly surface observations for ozone are taken from the European Environment Agency database (complied by Grange (2017)) and the US Environmental Protection Agency Air Quality
System (http://aqsdr1.epa.gov/aqsweb/aqstmp/airdata/download_files.html#Raw). Only "background" sites are considered in the analysis: For the USA this includes sites defined by the EPA as "suburban" and "rural"; For Europe this included sites categorized as "urban background", "background", and "rural" (see Figure A2).

## 4.2  Global burdens

Standard global metrics for evaluation of tropospheric chemistry models include the global burdens of tropospheric ozone,
CO, and OH (Table 1). The global annual average ozone burden in the G5NR-chem amounts to $348\,Tg$, consistent with the GEOS-Chem CTM and with the Tropospheric Ozone Assessment Report ($320 - 370\,Tg$; Hu et al., 2017; Young et al., 2018). The global burden of tropospheric CO of $294\,Tg$ is consistent with the GEOS-Chem CTM and on the low end of the observationally-based estimate of Gaubert et al. (2016). The global mean OH concentration is lower than in the GEOS-Chem CTM and more consistent with observational constraints (Prinn et al., 2005; Prather et al., 2012). The differences appear to be
mainly driven by differences in the meteorological data.

## 4.3  Free tropospheric evaluation: OMI, ozonesonde, SEAC[4]RS

Figure 3 compares the GEOS-5 Nature Run with GEOS-Chem chemistry to OMI mid-tropospheric ozone. GEOS-Chem CTM results from Hu et al. (2017) are also shown. OMI data have been reprocessed with a single fixed a priori profile (so that variability is solely due to observations), corrected for their global mean bias relative to ozonesondes, and filtered for high
latitudes because of large biases (Hu et al., 2017). Model outputs are sampled along the OMI tracks and smoothed with the OMI averaging kernels. The G5NR-chem captures well-known major features of the ozone distribution, such as ozone enhancements at northern mid-latitudes during MAM and JJA, and downwind of South America and Africa during SON. It shows no significant global bias relative to OMI and relative to the off-line GEOS-Chem CTM. The global mean seasonal



biases are less than $2.7 \pm 3.2\,\mathrm{ppbv}$. Spatial correlations for the four seasons on the $2° \times 2.5°$ grid scale are high and show no significant latitudinal bias (R=0.81-0.93; Figure 4).

Figure 5 further evaluates the simulated vertical distribution of ozone in comparison to ozonesonde data. There are differences between the G5NR-chem and the GEOS-Chem CTM that could be due to a number of factors including differences in

tropopause altitude and the distribution of lightning. For example, although the annual total lightning $NO_x$ emission in G5NR-chem is scaled to that in GEOS-Chem CTM, the lightning location is not constrained by satellite lightning flash data as the CTM does. Annual mean ozone biases in the G5NR-chem are generally less than $6\,\mathrm{ppbv}$ in the lower troposphere. There are some larger biases in the upper troposphere including differences with the GEOS-Chem CTM that could be due to the spatial distribution of lightning. Overall the G5NR-chem tends to improve the simulation of ozone vertical profiles compared to the

GEOS-Chem CTM, most dramatically at high southern latitudes.

Figure 6 compares the model to mean vertical profiles of ozone and precursors measured over the Southeast US during the SEAC[4]RS aircraft campaign. Here the GEOS-Chem CTM results are from a nested $0.25° \times 0.3125°$ simulation by Travis et al. (2016). The lower ozone in the northern mid-latitudes upper troposphere in the G5NR-chem appears to be due to a weaker lightning $NO_x$ source. The G5NR-chem overestimates ozone in the lower troposphere by $10\,\mathrm{ppbv}$, while such bias is reduced

in the GEOS-Chem CTM, even through both show almost identical $NO_x$ levels. The lower HCHO over the Southeast US is due to weaker isoprene emission because of lower temperatures. Both models underestimate CO in the free troposphere but the bias is more apparent in the CTM, likely due to differences in fire emissions, global OH fields, or transport error in off-line simulations (Yu et al., 2018).

### 4.4   Evaluation with surface observations over US and Europe

Figure 7 shows monthly median surface ozone concentrations grouped by regions in the US and Europe. Here hourly data between noon and 16:00 local time are used to remove the known issue that models typically underestimate the ozone nighttime depletion at surface sites (e.g., Millet et al., 2015). The G5NR-chem systematically overestimates surface ozone in almost all months by about $10\,\mathrm{ppbv}$ in all regions; while the GEOS-Chem CTM has no or small bias in winter and spring, but shows similar overestimates as in the G5NR-chem during summer and fall. In general the GEOS-5 Nature Run with GEOS-Chem

chemistry better captures the observed seasonality. Expanding the analysis to all hourly data does not affect the systematic bias in the G5NR-chem significantly, but tends to increases the summer/fall bias in the CTM (Figure A5). Part of the systematic bias is due to the subgrid vertical gradient between the lowest model level and the measurement altitude (60 m above ground vs 10 m, Travis et al., 2017). Recent model developments such as improved halogen chemistry and ozone dry deposition are expected to reduce the surface high bias (Schmidt et al., 2016; Sherwen et al., 2016a, b, 2017; Silva and Heald, 2018). These updates are

being incorporated into the GEOS-Chem version currently under development and will be passed on to the GEOS-5 simulation as they become available.



# 5 Conclusions

We presented a 1-year global simulation of tropospheric chemistry within the NASA GEOS ESM, version 5 (GEOS-5) at cubed-sphere c720 (~$12.5 \times 12.5 \, \mathrm{km}^2$) resolution. This demonstrated the success of implementing the GEOS-Chem chemical module within an ESM for on-line simulations with detailed chemistry. The GEOS-Chem chemical module on-line within GEOS and off-line as the GEOS-Chem CTM uses exactly the same code. In this way, the continual stream of chemical updates from the large GEOS-Chem CTM community can be seamlessly incorporated as updates to the on-line model, which always remains state-of-the-science and referenceable to the latest version of the GEOS-Chem. This 1-year simulation addressed an immediate need to generate the Nature Run for OSSEs in support of the geostationary satellite constellation for tropospheric chemistry. More broadly, implementation of the GEOS-Chem capability opens up a new capability for GEOS to address aerosol-chemistry-climate interactions, to assimilate satellite data of atmospheric composition, and to develop global air quality forecasts.

The 1-year GEOS-5 simulation at c720 resolution required 31 days walltime on 4705 cores. 45% of the wall time was spent on model dynamics and physics, 31% on input/output, and 24% on chemistry. Chemistry has near-perfect scalability in massively parallel architectures because it operates on individual grid columns, thus it is no longer a computing bottleneck in ESM simulations. Transporting the large number of species involved in atmospheric chemistry simulations may be a greater challenge.

We evaluated the GEOS-5 Nature Run with GEOS-Chem chemistry for consistency with the off-line GEOS-Chem CTM at coarser resolution ($2° \times 2.5°$ global and $0.25° \times 0.3125°$ nested over North America) as well as an ensemble of global observations for tropospheric ozone and aircraft observations of ozone precursors over the Southeast US. The model shows no significant global bias relative to OMI mid-tropospheric ozone data and the off-line GEOS-Chem CTM. Evaluations with ozonesondes show reduced model biases for high-latitude ozone. The GEOS-5 Nature Run with GEOS-Chem chemistry systematically overestimates surface ozone concentrations by $10 \, \mathrm{ppbv}$ all year round in the US and Europe, but is able to capture the observed seasonality; while the off-line GEOS-Chem CTM reproduces observed surface ozone levels in winter and spring, but has similar biases in summer and fall in all regions. Resolving this surface bias is presently a focus of attention in the GEOS-Chem CTM community and future model updates to address that bias can then be readily implemented into GEOS-5.

*Data availability.* All model outputs are available for download at https://portal.nccs.nasa.gov/datashare/G5NR-Chem/Heracles/12.5km/DATA or can be accessed through the OpenDAP framework at the portal https://opendap.nccs.nasa.gov/dods/OSSE/G5NR-Chem/Heracles/12.5km.

*Code availability.* GEOS-Chem CTM is avaiable at http://geos-chem.org/. GEOS-5 is available https://geos5.org/wiki/index.php?title=GEOS-5_public_AGCM_Documentation_and_Access.





*Acknowledgements.* This work was supported by the NASA Modeling, Analysis, and Prediction Program (MAP). Resources supporting the GEOS-5 simulations were provided by the NASA Center for Climate Simulation at Goddard Space Flight Center. LH acknowledges high-performance computing support from NCAR's Computational and Information Systems Laboratory, sponsored by the National Science Foundation. MJE and TMS acknowledge funding for the computational resource to perform the GEOS-Chem CTM model run from UK Nat-

5   ural Environment Research Council (NERC, BACCHUS project NE/L01291X/1). We also acknowledge all contributors to the ozonesonde data achieved at World Ozone and Ultraviolet Radiation Data Centre (WOUDC) website (http://www.woudc.org).



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



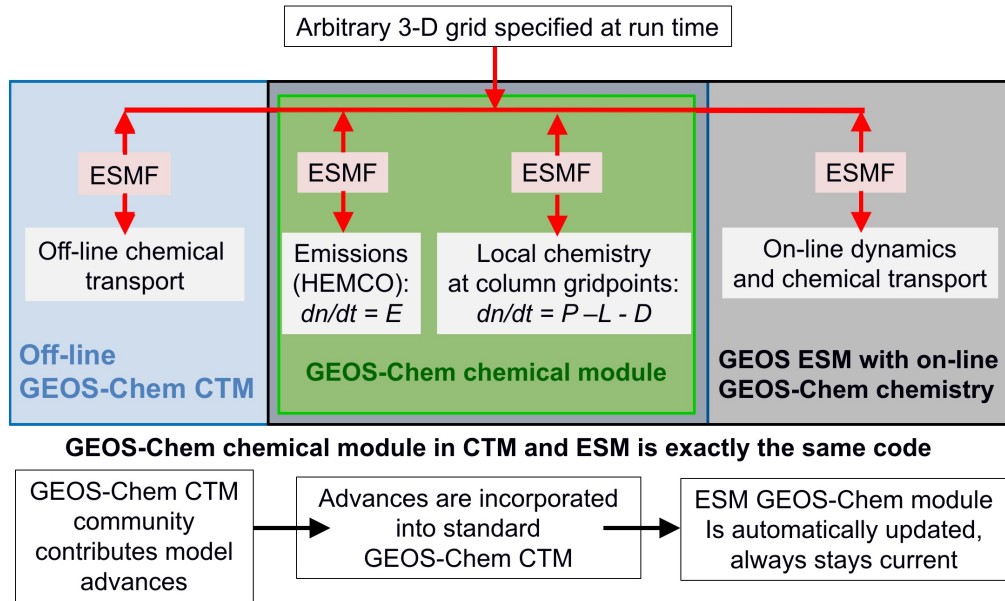

**Figure 1.** Schematic of GEOS-Chem chemical module used either off-line as a Chemical Transport Model (CTM) or on-line in an Earth System Model (ESM), with interfaces managed through the Earth System Modeling Framework (ESMF). The module is grid-independent with individual computations done on atmospheric columns at user-selected gridpoints. It computes local changes in concentrations with time ($dn/dt$) as a result of emissions ($E$), chemical production ($P$), chemical loss ($L$), and deposition ($D$). Emissions are handled through HEMCO to provide ESM users the option of only integrating emissions.





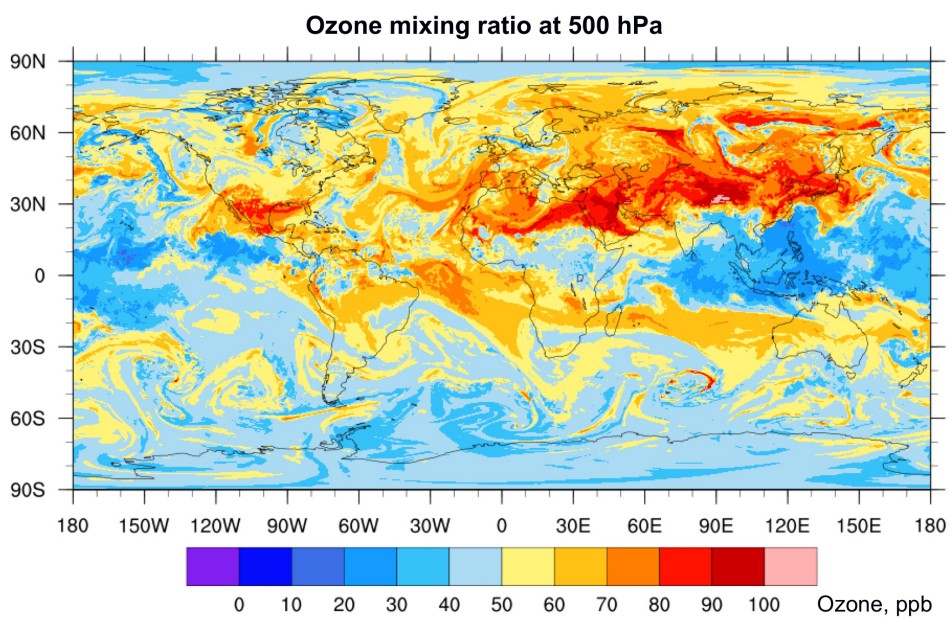

**Figure 2.** 500 hPa ozone distribution on August 1, 2013 at 0Z simulated by GEOS-5 with the GEOS-Chem chemical module at cubed-sphere c720 (~$12.5 \times 12.5\,\mathrm{km}^2$) resolution.



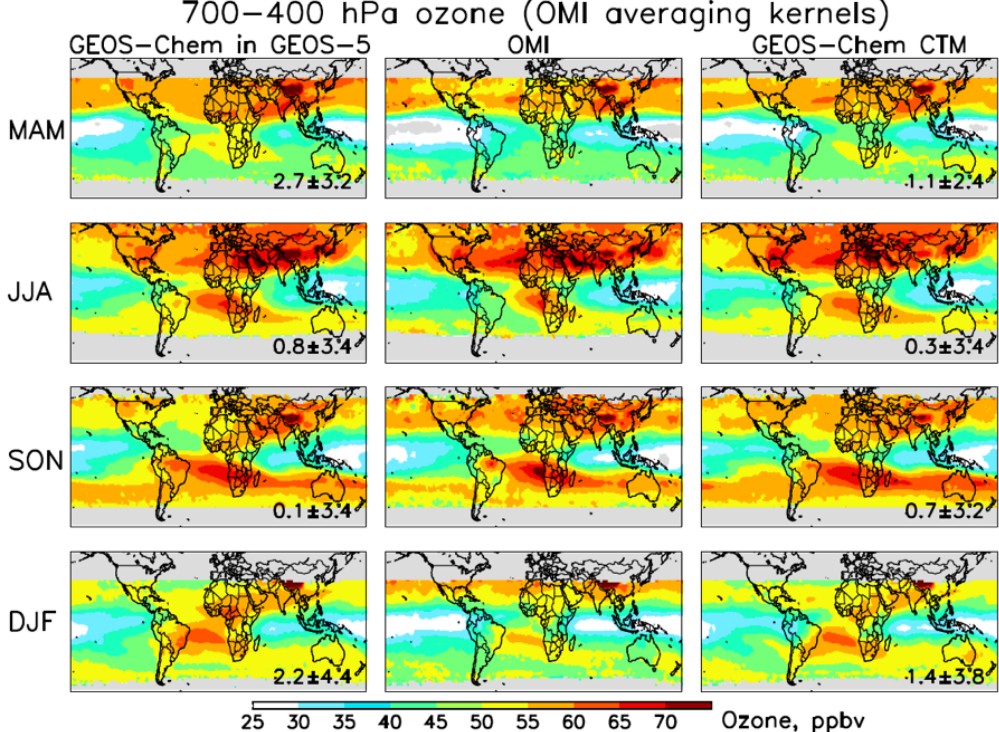

**Figure 3.** Middle tropospheric ozone distribution at 700-400 hPa from GEOS-5 Nature Run with GEOS-Chem chemistry (left column), OMI satellite observations (middle column), and the off-line GEOS-Chem CTM (right column) for four seasons covering July 2013 - June 2014. The data are on a $2° \times 2.5°$ grid. All data are smoothed by the OMI averaging kernels, using a single fixed a priori profile so that variability is solely driven by observations. The OMI observations have been further corrected for a global mean positive bias of 3.6 ppbv (Hu et al., 2017). Both models are sampled along the OMI tracks. Numbers in the left and right columns are the mean model bias ± standard deviation. Gray shading indicates regions where OMI data are unreliable and not used (poleward of 45° in winter-spring and poleward of 60° year-round; see Hu et al. (2017)).



**Figure 4.** Comparison of GEOS-5 Nature Run with GEOS-Chem chemistry to OMI 700-400 hPa ozone measurements for four seasons in July 2013 - June 2014, colored by the latitude of the observations. Each point represents the seasonal mean for a $2° \times 2.5°$ grid cell. Black dashed lines show the best fit (reduced major axis regression) with regression parameters given inset. Numbers on the bottom right are the global mean model bias ± standard deviation. The 1:1 line is shown in red.





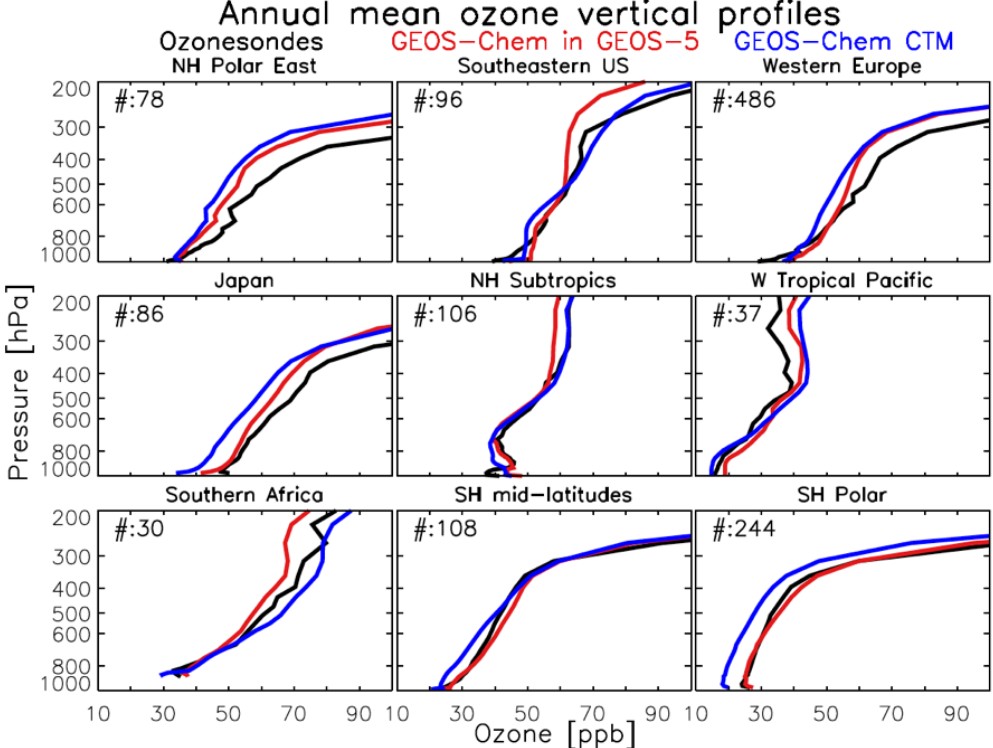

**Figure 5.** Annual mean ozonesonde profiles in July 2013 - June 2014 for representative global regions (Tilmes et al., 2012). Results from the GEOS-5 Nature Run with GEOS-Chem chemistry (red) are compared to observations (black) and to the GEOS-Chem CTM (blue; 2°×2.5° version in Hu et al. (2017)). The models are sampled at the ozonesonde launch times and locations.

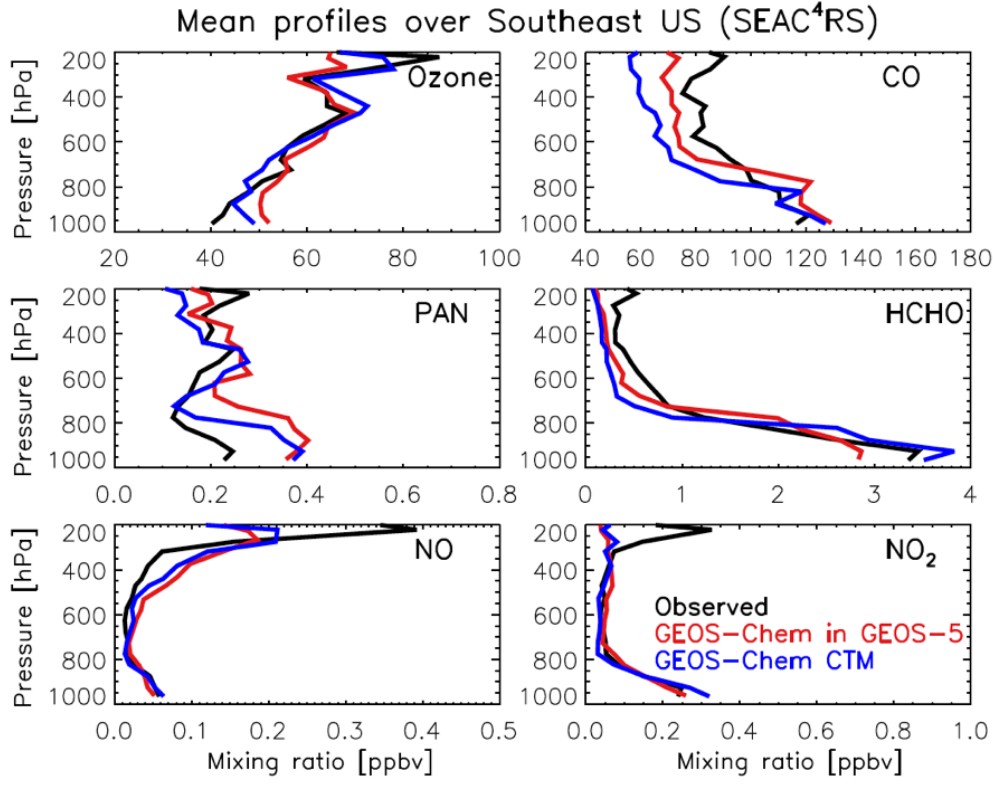

**Figure 6.** Mean vertical profiles of trace gas concentrations over the Southeast US during the NASA SEAC[4]RS aircraft campaign (Aug-Sep 2013, Toon et al. (2016)). Results from the GEOS-5 Nature Run with GEOS-Chem chemistry are compared to observations for ozone and $NO_x$, CO, peroxyacetyl nitrate (PAN), and formaldehyde (HCHO), and to the GEOS-Chem CTM (nested $0.25° \times 0.3125°$ version in Travis et al. (2016)). Model results are sampled along the flight tracks at the time of flights.





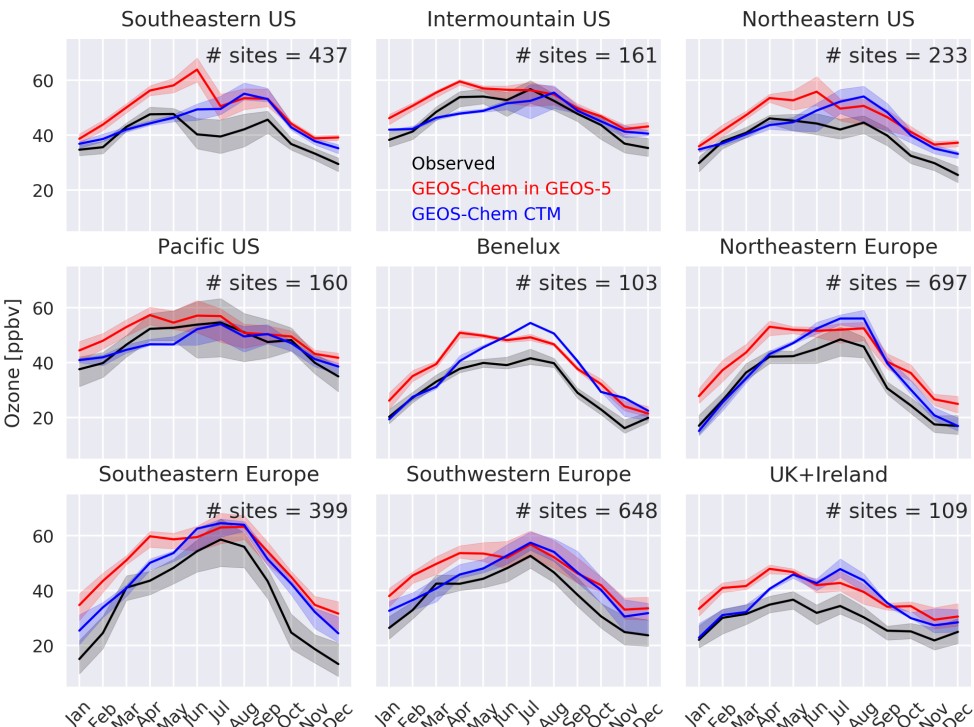

**Figure 7.** Monthly median afternoon (12-16 local time) ozone concentrations (ppbv) for 2013/2014 at surface sites in the US and Europe, with interquartile range shaded. Surface monitoring sites are grouped according to Figure A2. Also shown are the GEOS-5 Nature Run with GEOS-Chem chemistry (red) and the off-line GEOS-Chem CTM (blue). Hourly model outputs are sampled for the locations and time of observations at the surface (lowest) level.



**Table 1.** Global tropospheric burdens.

| | GEOS-5 Nature Run with GEOS-Chem chemistry | GEOS-Chem CTM [a] | Literature range |
|---|---|---|---|
| O$_3$ burden [Tg] | 348 [b] | 347 [b] | 320-370 [c] |
| CO burden [Tg] | 294 | 285 | 290-370 [d] |
| mean OH [$1 \times 10^5$ molecule cm$^{-3}$] [e] | 10.2 | 12.5 | 9.5-12.7 [f] |

[a] Hu et al. (2017).

[b] Calculated with a chemical tropopause as the 150 ppbv ozone isopleth.

[c] Interquartile range of 50 models summarized in Young et al. (2018); limited observational estimates fall within that range.

[d] Gaubert et al. (2016).

[e] Global mean airmass-weighted OH concentration.

[f] From 16 model results summarized in Naik et al. (2013)





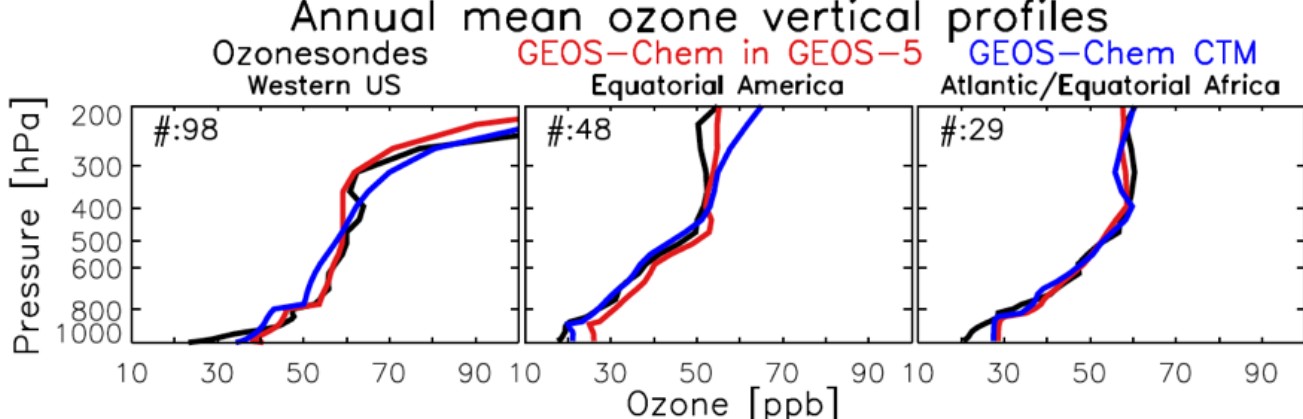

**Figure A1.** Same as Figure 5 but for Western US, Equatorial American, and Atlantic/Equatorial Africa.

**Appendix A**



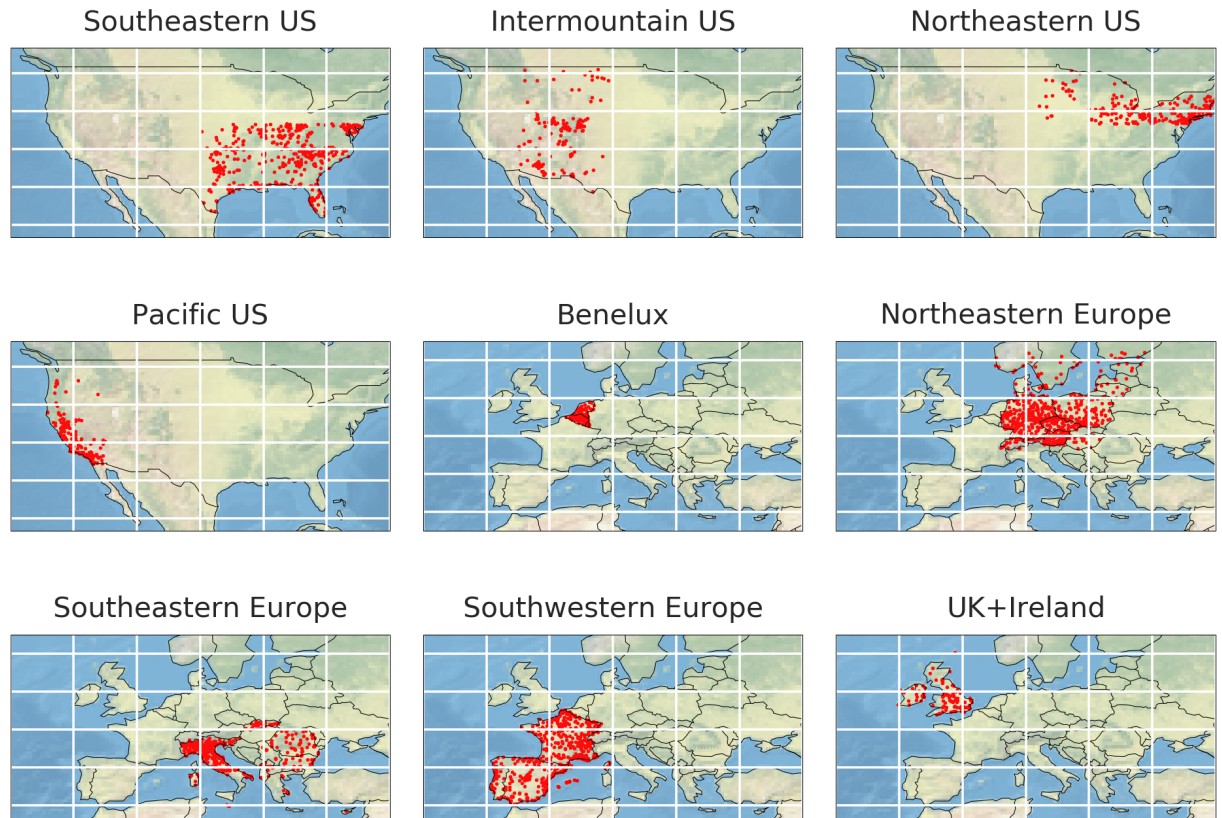

**Figure A2.** Surface monitoring sites in the US and Europe grouped by subregions as analyzed in the text. Background image ©Natural Earth (public domain license)



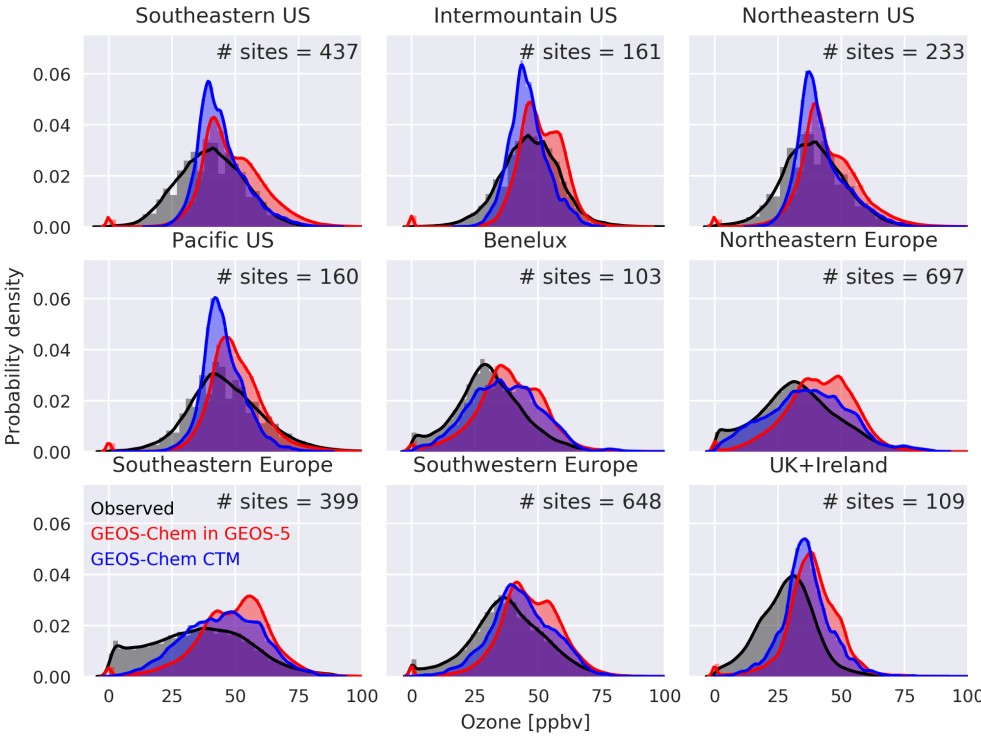

**Figure A3.** Probability distribution of afternoon (12-16 local time) ozone concentrations (ppbv) for 2013/2014 at surface sites in the US and Europe.





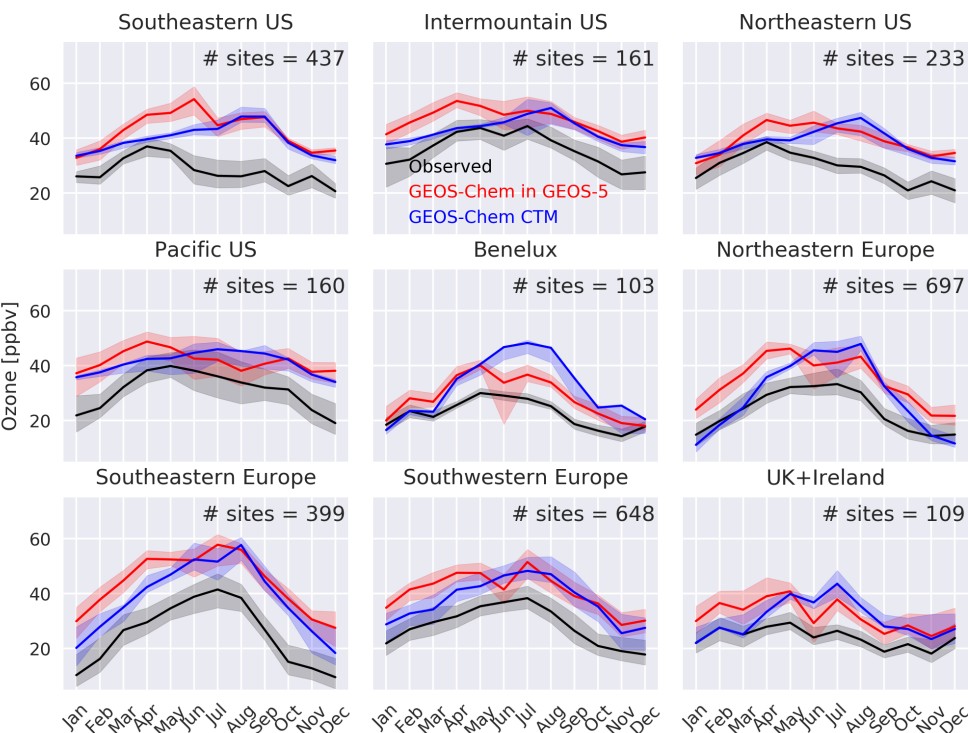

**Figure A4.** Monthly median ozone concentrations (ppbv; all data) for 2013/2014 at surface sites in the US and Europe, with interquartile range shaded. Surface monitoring sites are grouped according to Figure A2. Also shown are the GEOS-5 Nature Run with GEOS-Chem chemistry (red) and the off-line GEOS-Chem CTM (blue). Hourly model outputs are sampled for the locations and time of observations at the surface (lowest) level.





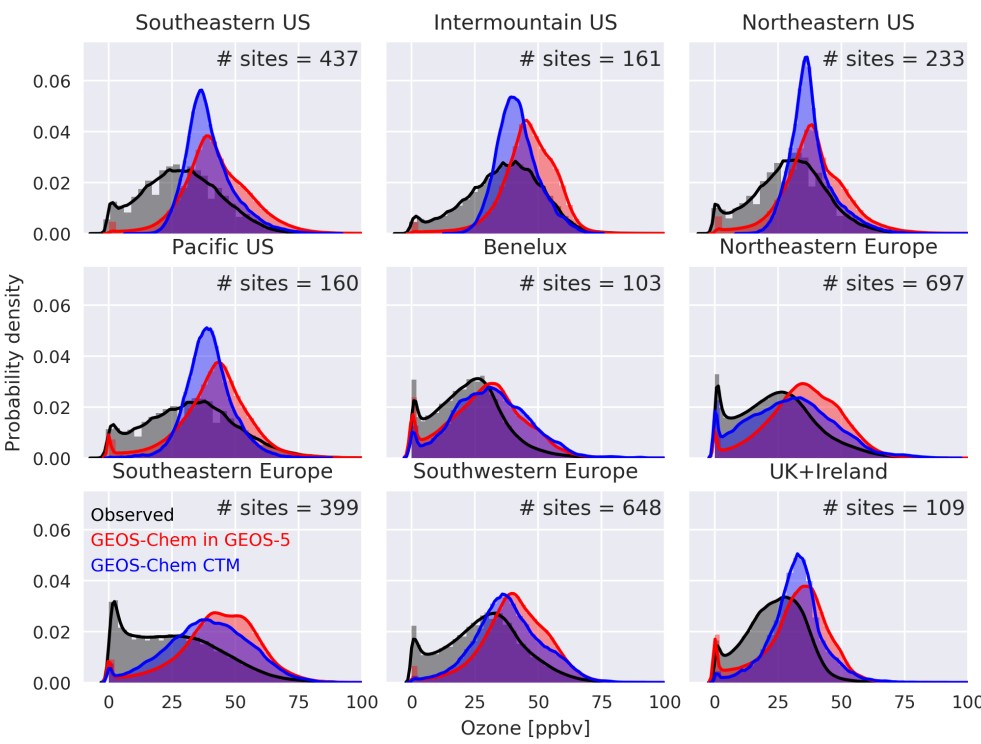

**Figure A5.** Probability distribution of ozone concentrations (ppbv; all data) for 2013/2014 at surface sites in the US and Europe.





**Table A1.** Computing resources for the GEOS-5 Nature Run with GEOS-Chem chemistry [a]

|  | Wall time breakdown | Wall time | Compute cores | Disk storage |
|---|---|---|---|---|
| Transport | 45% | 31 days | 4705 cores [b] | 180 Tb [c] |
| Chemistry | 24% | | | |
| Input | 25% | | | |
| Output | 6% | | | |

[a] The simulation is from July 1 2013 - July 1 2014 in GEOS ESM with GEOS-Chem as a chemical module at $12.5 \times 12.5 \, \text{km}^2$. The computation was carried out at NASA Discover supercomputing cluster.

[b] The simulation uses 337 14-core 2.6 GHz Intel Xeon Haswell compute nodes with 128 GB of memory per node, and an Infiniband FDR interconnect using the Intel compiler suite (v. 15.0.0.090) and MPT v. 2.11.

[c] 158 species are simulated and transported by the GEOS ESM; among them 29 species are saved out as hourly outputs. Data is available at
https://portal.nccs.nasa.gov/datashare/G5NR-Chem/Heracles/12.5km/DATA/

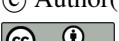

**Table A2.** Emissions in the GEOS-5 Nature Run with GEOS-Chem chemistry

| Species | EDGAR HTAP v2 [h] | RETRO [i] | Anthropogenic Xiao et al. (2008)[a] | NEI 2011 [j] | MIX [k] | Aircraft AEIC [l] | Ship HTAP [h] | Volcanoes [b] | Biomass burning QFED [m] | Biogenic MEGAN [n] | Soil & agriculture [c] | Lightning [d] | Oceanic/Sea spray [e] | Dust [f] | Annual emission (Tg/y or TgC/y [g]) |
|---|---|---|---|---|---|---|---|---|---|---|---|---|---|---|---|
| NO | X | | | X | | X | X | | X | | X | X | | | 117 |
| CO | X | | | X | | X | X | | X | X | | | | | 958 |
| SO₂ | X | | | X | | X | X | X | X | | | | | | 129 |
| SO₄ | X | | | X | | X | | | | | | | | | 2.93 |
| Organic carbon | X | | | X | | X | X | | X | X | | | | | 72.1 |
| Black carbon | X | | | X | | X | X | | X | | | | | | 10.3 |
| NH₃ | X | | | X | | | | | | | X | | | | 60.5 |
| ≥C₄ alkanes | | X | | X | X | X | | | | | | | | | 32.1 |
| Acetone | | X | | X | X | X | | | X | X | | | X | | 77.6 |
| MEK | | X | | X | X | | | | | | | | | | 4.09 |
| CH₃CHO | | X | | X | X | X | | | | X | | | | | 20.9 |
| C₃H₆ | | X | | X | X | X | | | | X | | | | | 31.3 |
| C₃H₈ | | X | | X | X | X | | | | | | | | | 6.33 |
| CH₂O | | X | | X | X | X | | | | | | | | | 4.58 |
| Isoprene | | | | | | | | | | X | | | | | 345 |
| C₂H₆ | | | X | | | X | | | | | | | | | 15.3 |
| CHBr₃ | | | | | | | | | | | | | X | | 0.429 |
| CH₂Br₂ | | | | | | | | | | | | | X | | 0.0621 |
| Br₂ | | | | | | | | | | | | | X | | 0.63 |
| Sea salt A (0.1 − 0.5μg) | | | | | | | | | | | | | X | | 65.9 |
| Sea salt C (0.5 − 4.0μg) | | | | | | | | | | | | | X | | 3990 |
| DMS | | | | | | | | | | | | | X | | 35.2 |
| Dust1 (0.1 − 1.0μg) | | | | | | | | | | | | | | X | 103 |
| Dust2 (1.0 − 1.8μg) | | | | | | | | | | | | | | X | 212 |
| Dust3 (1.8 − 3.0μg) | | | | | | | | | | | | | | X | 271 |
| Dust4 (3.0 − 6.0μg) | | | | | | | | | | | | | | X | 254 |

[a] Excluding aircraft and ships, which are listed separately.

[b] http://aerocom.met.no/download/emissions/AEROCOM_HC/volc/.

[c] Hudman et al. (2012).

[d] Murray et al. (2012).

[e] Jaeglé et al. (2011) and Fischer et al. (2012).

[f] Zender et al. (2003).

[g] TgC/y for ≥C₄ alkanes, Acetone, MEK, CH₃CHO, C₃H₆, C₃H₈, Isoprene, and C₂H₆. Tg/y for the rest species.

[h] http://edgar.jrc.ec.europa.eu/htap_v2/index.php?SECURE=123

[i] RETRO monthly global inventory for 2000 (Schultz et al., 2007) implemented as described by Hu et al. (2015a).

[j] US EPA National Emission Inventory 2008 http://www.epa.gov/ttn/chief/net/2008report.pdf and scaled to 2013 https://www3.epa.gov/airtrends/.

[k] Asian anthropogenic emissions (Li et al., 2014).

[l] Stettler et al. (2011).

[m] Quick Fire Emissions Dataset (QFED) version 2.4-r6 (Darmenov and Da Silva, 2015).

[n] MEGANv2.1 (Guenther et al., 2012) implemented in GEOS-Chem as described by Hu et al. (2015b).