# Peer review of "Global simulation of tropospheric chemistry at 12.5 km resolution: performance and evaluation of the GEOS-Chem chemical module (v10-1) within the NASA GEOS Earth System Model (GEOS-5 ESM)"

_Geoscientific Model Development, 2018_

## Referee Comment (RC1) · Anonymous Referee #1 · 7 Jun 2018

General Comments:

This paper presents an online version of the GEOS-chem chemistry module with an aim of simulating tropospheric chemistry at very high horizontal resolution for a global model (appox. 12.5km). More imporantly the authors have developed a consistency between the GEOS-chem CTM and the online ESM which enables the ESM to keep up with the state of the art science. The main aim is to serve observing simulation experi-

ments for satellite monitoring systems but also has the added advantage of being able to simulate high-resolution impacts of climate change on air quality. The high resolution of the output will also be of interest to the health and vegetation impacts communities. The paper is very well written and easy to follow and will be of great interest to the wider atmospheric science community. Therefore I recommend publication in GMD after the following relatively minor comments are addressed.

Specific Comments:

Page 5, Line 9: You mention that the model has 72 vertical levels here up to a height of 0.01hPa. Would you be able to clarify what the spacing of the lower vertical model levels are, specifically in the boundary layer region? This will be important if one of the intended uses of the model is for air quality simulations as can have important implications for the rise of emissions plumes, mixing of emissions throughout the BL and ultimately concentrations of key secondary species such as ozone. Further to this the spacing of vertical layers near to the surface can also impact which height emissions are injected at which could have implications for ozone destruction pathways (E.g. NOx titration) and thus concentrations. This could be one of the potential reasons why the model is failing to capture the surface concentrations in Europe (although it is doing a reasonably good job at present). I think the discussion of model performance could benefit further from some extra discussion in relation to this matter.

Page 5, Line 25: Please provide a brief summary of the VOCs included. From an air quality perspective it would be interesting to know the most reactive VOCs from an ozone formation potential. Higher reactive VOCs (E.g. Butanes and aromatics) are important for looking at air quality issues over highly polluted regions such as northern China where models with more basic chemistry schemes struggle to reproduce high levels of O3.

Page 5, Line 26: Which aerosol microphysics scheme is being used?

Page 5, Line 27: Is Fast-JX fully coupled to cloud and aerosol species, to simulate

feedbacks? This could have implications on chemical processes. This is especially important given that the focus of evaluation is ozone.

Technical corrections:

Page 3, Line 22: Double period at the end of the sentence please remove.

Page 4: Lines 32-33: Which boundary layer scheme is being used here? Please provide some detail.

Page 7, Lines 23 and 24: The observed/literature values for O3 are shown in the text and Table 1 but the values for CO and OH are only shown in Table 1. Please be consistent and either refer to all observed values in the text or remove the mention to the O3 values in the text and refer to Table 1.

---

## Referee Comment (RC2) · Anonymous Referee #2 · 23 Jul 2018

Review of GMDD 2018-111

This paper presents and extraordinary and highly valuable piece of work. It is well written and succinct, most unusual given the scope of work involved. It is publishable as is, but given the unique nature of developing a 12.5-km gridded chemistry run, the authors should add just a bit more of information to address some of the obvious questions.

(1) Comment on the lack of improvement in vertical resolution (L72) while jumping up the horizontal. For example, ECMWF now uses L91 or L137 with high-res models.

(2) You talk of a 300 s heartbeat step across the physics and dynamics, but what is the exchange time with the chemistry and how fast is the chemistry updated for tracer transport? There is some confusion in Sect 3.1 about the chemistry time step.

(3) As one gets to 10 km, there are a number of issues that arise in our standard treatment of the continuity equations. Presumably the large-scale transport adjust u,v,w such that the hybrid coordinate system is maintained when it passes to the chemistry? This creates a problem when doing convection and scavenging of tracers in the chemistry package because convective mass fluxes are balanced by u,v flows – especially when one gets to 10 km scales. If the chemistry package does convective upward fluxes and then balances with downward flow to balance the coordinate system, then there is a false downward transport in the column, and the results will have increasing errors as the resolution gets smaller. This really should be discussed, explained what is actually done (this is not here), and then assess possible errors from the formulation. To be correct one has to have the large-scale and convective w's done together, since convection involves neighboring columns. For example, some CTMs allow the u,v step to leave the air mass in each layer that may not be according to the hybrid coords and then the convection in the single column corrects it back to the std levels. If you did this typical operator split, then say so. If not, note it.

(4) The approach taken here for assessing the surface ozone (air quality) simulation is very disappointing and this is the only area that needs a substantial redo. First, they build a 12 km model and then go back to the old-fashioned way of correcting for bad simulations of air quality that may be needed in 2 deg models (i.e., only compare with 4 pm values). Further, with G5NR-chem there should be no need to compare only with "background sites" – the polluted regions should be included in the comparisons. (There are reasons to remove of the "traffic sites" in the EU data, but eliminating all urban data gives you the wrong comparison. The idea that you compare with only

afternoon ozone dates back to models that could not really do realistic surface ozone – this model should be able to compare with the true AQ data and for all hours. You have all the simulated data (since this is save for an OSSE) and so it is straightforward and essential that a more serious and more accurate comparison with surface ozone be done. There are numerous better publications, including multi-model evaluations, please start with one of those. For example, you could be using the Schnell gridded surface ozone and doing the comparisons as in the Schnell 2015 ACP paper Figure 1 – It would be good to see how the seasonal and diurnal cycles differ between the 2 chemistry models. Figures A3 and A5 are very interesting – presumably this includes all 24-hr data?

(5) The concept of tropospheric burden depends a lot on the resolution of the tropopause structures. It would be very useful to discuss (show may be too much here) how the G5NR-Chem model reproduces the folds around (above and below) the jet where the STE takes place. Is the tropospheric mass the same in both chem models at 0.1 and 2.0 degrees? STE flux is a very important model diagnostic, and since you are running Linoz in both models it is trivial to evaluate from the flux needed (presumably archived) to reset the O3strat in the boundary layer. Such a simple monthly, hemispheric value would help greatly in understanding differences between the two models.

(6) Another problem with increased resolution is the need to connect neighboring columns for the radiation and photolysis. In terms of radiation, the direct sunlight in one column often passes through and is scattered or absorbed by a neighboring columns. The overhead ozone column is often not the effective ozone column. The effect of neighboring (unknown) columns is very important for clouds. I doubt that the model is able to considers this effect, but it should be noted as this may be a case where increasing resolution may actually be worse. Related to this, it is also important to described how the Fast-J code treats clouds and aerosols. This later is an easy fix.

(7) The authors have not really shown off the capability of a 12-km global chemistry

model. Particularly disappointing the comparisons with data from sondes or surface sites where the resolution of the G5NR is dumbed-down to compare with the GEOS-chem reference run at 222 km [I think this is what saw]. It is important to see what the 222 km result from G5NR looks like when comparing with GEOSchem, but then to see the better structures and better match when using 12-km results.

Minor: p.6 l.22: 'a . . . OSSEs . . .' singular or plural? p.9 l29: what is code availability for G5NR-chem?

––––––––––––––––––––––––––––––––––

---

## Author Comment (AC1) · 4 Sep 2018

**Response**: We appreciate the positive comments from the Anonymous Referee #1. We further clarified the manuscript following the reviewer's suggestions. Reviews are in red, while responses are in black.

General Comments:
This paper presents an online version of the GEOS-chem chemistry module with an aim of simulating tropospheric chemistry at very high horizontal resolution for a global model (appox. 12.5km). More importantly the authors have developed a consistency between the GEOS-chem CTM and the online ESM which enables the ESM to keep up with the state of the art science. The main aim is to serve observing simulation experiments for satellite monitoring systems but also has the added advantage of being able to simulate high-resolution impacts of climate change on air quality. The high resolution of the output will also be of interest to the health and vegetation impacts communities. The paper is very well written and easy to follow and will be of great interest to the wider atmospheric science community. Therefore I recommend publication in GMD after the following relatively minor comments are addressed.

Specific Comments:
Page 5, Line 9: You mention that the model has 72 vertical levels here up to a height of 0.01hPa. Would you be able to clarify what the spacing of the lower vertical model levels are, specifically in the boundary layer region? This will be important if one of the intended uses of the model is for air quality simulations as can have important implications for the rise of emissions plumes, mixing of emissions throughout the BL and ultimately concentrations of key secondary species such as ozone. Further to this the spacing of vertical layers near to the surface can also impact which height emissions are injected at which could have implications for ozone destruction pathways (E.g. NOx titration) and thus concentrations. This could be one of the potential reasons why the model is failing to capture the surface concentrations in Europe (although it is doing a reasonably good job at present). I think the discussion of model performance could benefit further from some extra discussion in relation to this matter. Page 5, Line 25: Please provide a brief summary of the VOCs included. From an air quality perspective it would be interesting to know the most reactive VOCs from an ozone formation potential. Higher reactive VOCs (E.g. Butanes and aromatics) are important for looking at air quality issues over highly polluted regions such as northern China where models with more basic chemistry schemes struggle to reproduce high levels of O3.
**Response**: The vertical model levels can be found at the GEOS-Chem Wiki page (http://wiki.seas.harvard.edu/geos-chem/index.php/GEOS-Chem_vertical_grids#72-layer_vertical_grid). Specifically, 14 layers are below 2 km altitude, with spacing spanning from 64 m to 128 m and the lowest layer from the surface to 58 m above ground. There are 15 layers between 2 km and 10 km; the rest 43 layers between 10 and the top of atmosphere (80 km). Developing higher vertical resolution with more layers in the boundary layer and near tropopause is ongoing work at the NASA GMAO; thus, when ready it will benefit both GEOS and GEOS-Chem models.

Anthropogenic emissions are distributed vertically according to the corresponding emission inventories when information is available. We agree with the reviewer that reasons for model ozone bias could be related to vertical distribution for emission and lowest model layer height,

thus it was mentioned in the manuscript that 'Part of the systematic bias is due to the subgrid vertical gradient between the lowest model level and the measurement altitude (60 m above ground vs. 10 m, Travis et al., 2017).'

Same as GEOS-Chem, the G5NR-chem includes the following non-methane VOCs with direct emissions: isoprene, $\geq C_4$ alkanes, $\geq C_4$ ketones, propane, acetone, $\geq C_3$ alkenes, ethane, ethene, acetaldehyde, formaldehyde, and this is summarized by Hu et al 2017 and cited in the manuscript. Aromatic chemistry is not included in this version of the models, but is being under development for the future GEOS-Chem version (Yan et al., 2018). This will be passed on to the future GEOS-5 simulation when available. We added this into the discussion on the model ozone bias.

Page 5, Line 26: Which aerosol microphysics scheme is being used?
**Response**: The regular GEOS-Chem aerosol scheme TOMAS is used here. We note that in the G5NR-chem configuration, GEOS-Chem is run 'passively', i.e. the aerosols and trace gases do not influence the meteorology.  We clarified it in the revised manuscript.

Page 5, Line 27: Is Fast-JX fully coupled to cloud and aerosol species, to simulate feedbacks? This could have implications on chemical processes. This is especially important given that the focus of evaluation is ozone.
**Response**: See above. GEOS-Chem tracers are not impacting the meteorology. The primary purpose of the simulation was to evaluate newly developed GEOS-5 ESM with GEOS-Chem chemical model at very high resolution, without the additional complication of chemistry-climate feedbacks that are not included in a CTM. We clarified it in the revised manuscript.

Technical corrections:
Page 3, Line 22: Double period at the end of the sentence please remove.
**Response**: Thanks for pointing this. We corrected it in the revised manuscript.

Page 4: Lines 32-33: Which boundary layer scheme is being used here? Please provide some detail.
**Response**: We use the GEOS-5 boundary layer scheme, which uses a combination of Lock et al. (2000) and Louis and Geleyn (1982).

Page 7, Lines 23 and 24: The observed/literature values for O3 are shown in the text and Table 1 but the values for CO and OH are only shown in Table 1. Please be consistent and either refer to all observed values in the text or remove the mention to the O3 values in the text and refer to Table 1.
**Response**: Done.

**References:**
Lock, A. P., Brown, A. R., Bush, M. R., Martin, G. M., and Smith, R. N. B.: A New Boundary Layer Mixing Scheme. Part I: Scheme Description and Single-Column Model Tests, Monthly Weather Review, 128, 3187–3199, https://doi.org/10.1175/1520-0493(2000)128<3187:ANBLMS>2.0.CO;2, 2000

Louis, J.-F., Tiedtke, M., and Geleyn, J.-F.: A short history of the PBL parameterization at ECMWF, in: Workshop on Planetary Boundary Layer parameterization, 25-27 November 1981, pp. 59–79, ECMWF, ECMWF, Shinfield Park, Reading, 1982

Yan, Y., Cabrera-Perez, D., Lin, J., Pozzer, A., Hu, L., Millet, D. B., Porter, W. C., and Lelieveld, J.: Global tropospheric effects of aromatic chemistry with the SAPRC-11 mechanism implemented in GEOS-Chem, Geosci. Model Dev. Discuss., https://doi.org/10.5194/gmd-2018-196, in review, 2018.

**Review 2**
Review of GMDD 2018-111
This paper presents and extraordinary and highly valuable piece of work. It is well written and succinct, most unusual given the scope of work involved. It is publishable as is, but given the unique nature of developing a 12.5-km gridded chemistry run, the authors should add just a bit more of information to address some of the obvious questions.
**Response**: We appreciate the reviewer #2's positive feedback. We further clarified the manuscript following the reviewer's comments.

(1) Comment on the lack of improvement in vertical resolution (L72) while jumping up the horizontal. For example, ECMWF now uses L91 or L137 with high-res models.
**Response**: See the response to the review 1. Increasing vertical resolution is an ongoing work for improving the NASA GEOS ESM.

(2) You talk of a 300 s heartbeat step across the physics and dynamics, but what is the exchange time with the chemistry and how fast is the chemistry updated for tracer transport? There is some confusion in Sect 3.1 about the chemistry time step.
**Response**: Correct. Chemistry is done every 300s, and the chemical tracer fields are exchanged with all other components at the heartbeat. We clarified it in the revisions.

(3) As one gets to 10 km, there are a number of issues that arise in our standard treatment of the continuity equations. Presumably the large-scale transport adjust u,v,w such that the hybrid coordinate system is maintained when it passes to the chemistry? This creates a problem when doing convection and scavenging of tracers in the chemistry package because convective mass fluxes are balanced by u,v flows – especially when one gets to 10 km scales. If the chemistry package does convective upward fluxes and then balances with downward flow to balance the coordinate system, then there is a false downward transport in the column, and the results will have increasing errors as the resolution gets smaller. This really should be discussed, explained what is actually done (this is not here), and then assess possible errors from the formulation. To be correct one has to have the large-scale and convective w's done together, since convection involves neighboring columns. For example, some CTMs allow the u,v step to leave the air mass in each layer that may not be according to the hybrid coords and then the convection in the single column corrects it back to the std levels. If you did this typical operator split, then say so. If not, note it.
**Response**: The GEOS-Chem chemical module is fully embedded in GEOS-5 ESM, sharing the same cube-sphere and vertical coordinates at all times. We further added a discussion on the impact of resolution on convection in the revised manuscript: "Radon-222 tracer simulation tests

within GEOS-5 show that the GEOS-Chem convection scheme closely reproduces the GEOS-5 convective transport (Yu et al., 2018). As convection becomes increasingly resolved at higher model resolutions, the GEOS-5 sub-grid convection parameterization (Moorthi and Suarez, 1992) is invoked less frequently. As a consequence, an increasing fraction of the washout in GEOS-Chem becomes characterized as large-scale, as opposed to convective. No attempts were made to offset the possible increase in washout efficiency that may arise from this."

(4) The approach taken here for assessing the surface ozone (air quality) simulation is very disappointing and this is the only area that needs a substantial redo. First, they build a 12 km model and then go back to the old-fashioned way of correcting for bad simulations of air quality that may be needed in 2 deg models (i.e., only compare with 4 pm values). Further, with G5NR-chem there should be no need to compare only with "background sites" – the polluted regions should be included in the comparisons. (There are reasons to remove of the "traffic sites" in the EU data, but eliminating all urban data gives you the wrong comparison. The idea that you compare with only afternoon ozone dates back to models that could not really do realistic surface ozone– this model should be able to compare with the true AQ data and for all hours. You have all the simulated data (since this is save for an OSSE) and so it is straightforward and essential that a more serious and more accurate comparison with surface ozone be done. There are numerous better publications, including multi-model evaluations, please start with one of those. For example, you could be using the Schnell gridded surface ozone and doing the comparisons as in the Schnell 2015 ACP paper Figure 1 – It would be good to see how the seasonal and diurnal cycles differ between the 2 chemistry models. Figures A3 and A5 are very interesting – presumably this includes all 24-hr data?

**Response**: We appreciate the reviewer's perspectives. However, even at ~12.5km it is unrealistic to expect that sites with large local influence (e.g. roadside, city center, or heavily trafficked sites) would be captured. As many sites classified by European and USA data providers as without large local influence were therefore included. Thus as stated in the manuscript (Page 7 lines 18-22), "urban background" and "rural background" sites were used for Europe and "suburban" and "rural" for USA locations. It is worth stating that analysis here was limited by the supporting data for observations (ie. descriptions of site locations), which is hoped will be ameliorated in future with the availability of datasets with enhanced supporting data (such as the Tropospheric Ozone Assessment Report (TOAR) dataset - Schultz et al 2017).

As the review notes, gridded datasets exist for comparisons with models. However, these are at coarser resolution (up to 1x1 degree resolution; Sofen et al 2016, Schnell et al 2015) than the "Nature run" presented here. The preference was therefore to compare to point observations, rather than degrading the resolution of the "Nature run" for comparison against existing datasets.

As stated in the manuscript, models tend to have troubles to simulate diurnal cycles of ozone particularly at night, and we don't expect the "Nature run" would solve all model issues. Indeed, even regional air quality models at higher resolution cannot reproduce the diurnal cycle (e.g., Herwehe et al., 2011; Solazzo et al., 2012).

(5) The concept of tropospheric burden depends a lot on the resolution of the tropopause structures. It would be very useful to discuss (show may be too much here) how the G5NR-Chem model reproduces the folds around (above and below) the jet where the STE takes place.

Is the tropospheric mass the same in both chem models at 0.1 and 2.0 degrees? STE flux is a very important model diagnostic, and since you are running Linoz in both models it is trivial to evaluate from the flux needed (presumably archived) to reset the O3strat in the boundary layer. Such a simple monthly, hemispheric value would help greatly in understanding differences between the two models.

**Response**: Without the individual tendency terms (transport, washout, chemistry) it is difficult to evaluate the STE. The diagnostic O3strat as the reviewer suggested was not available when the Nature run was carried out, thus we are not able the access the STE flux directly. We try to minimize the impact of the tropopause by using the ozonopause in the burden calculations as the boundary between the troposphere and stratosphere. We also note that Knowland et al (2017) have demonstrated that the GEOS has skills in reproducing stratospheric intrusions. The main focus on the evaluation of tropospheric ozone (particularly through comparisons with ozonesonde data in the upper troposphere) does not suggest any significant discrepancy in STE. Indeed, the G5NR-Chem does a better job of simulating the upper tropospheric ozone than the GEOS-Chem CTM.

(6) Another problem with increased resolution is the need to connect neighboring columns for the radiation and photolysis. In terms of radiation, the direct sunlight in one column often passes through and is scattered or absorbed by a neighboring columns. The overhead ozone column is often not the effective ozone column. The effect of neighboring (unknown) columns is very important for clouds. I doubt that the model is able to considers this effect, but it should be noted as this may be a case where increasing resolution may actually be worse. Related to this, it is also important to described how the Fast-J code treats clouds and aerosols. This later is an easy fix.

**Response**: Thank you for pointing out these. We use the overhead column and don't consider the neighboring columns. For Fast-JX we use the approximate randomized cloud overlap method. The reviewer is correct that the high-res model may be worse here than the 2x25 degree CTM model, especially at high Solar Zenith Angles.

(7) The authors have not really shown off the capability of a 12-km global chemistry model. Particularly disappointing the comparisons with data from sondes or surface sites where the resolution of the G5NR is dumbed-down to compare with the GEOSchem reference run at 222 km [I think this is what saw]. It is important to see what the 222 km result from G5NR looks like when comparing with GEOSchem, but then to see the better structures and better match when using 12-km results.

**Response**: As an important model development as the implementation of a new chemical module into GEOS ESM, examining the consistency between the GEOS-Chem chemical module within the GEOS ESM and the off-line GEOS-Chem CTM is a critical step. Differences between the G5NR-chem and the CTM cannot be solely attributed to the different resolution, but should also be due to differences in transport, meteorological data, etc. The impact of high resolution on atmospheric chemistry and air quality would be better addressed with the GEOS-5 ESM run at the different resolution, which will be investigated in the future.

Minor: p.6 l.22: 'a : : : OSSEs : : :' singular or plural? p.9 l29: what is code availability for G5NR-chem?

**Response**: Done. G5NR-chem is the model output simulated by the GEOS-5 ESM with the GEOS-Chem, thus GEOS-5 code availability is reported while G5NR-chem data availability is reported.

**References**:
Herwehe, J. A., Otte, T. L., Mathur, R., & Rao, S. T. (2011). Diagnostic analysis of ozone concentrations simulated by two regional-scale air quality models. Atmospheric Environment, 45(33), 5957–5969. https://doi.org/10.1016/j.atmosenv.2011.08.011

Knowland, K. E., Ott, L. E., Duncan, B. N., & Wargan, K. (2017). Stratospheric intrusion-influenced ozone air quality exceedances investigated in the NASA MERRA-2 reanalysis. *Geophysical Research Letters*, 44, 10,691–10,701. https://doi.org/10.1002/2017GL074532

Schultz, M. G. and 96 co-authors (2017), Tropospheric Ozone Assessment Report: Database and metrics data of global surface ozone observations, Elem Sci. Anth, 5, DOI: http://doi.org/10.1525/elementa.244

Schnell, J. L., Prather, M. J., Josse, B., Naik, V., Horowitz, L. W., Cameron-Smith, P., Bergmann, D., Zeng, G., Plummer, D. A., Sudo, K., Nagashima, T., Shindell, D. T., Faluvegi, G., and Strode, S. A.: Use of North American and European air quality networks to evaluate global chemistry–climate modeling of surface ozone, Atmos. Chem. Phys., 15, 10581-10596, https://doi.org/10.5194/acp-15-10581-2015, 2015.

Solazzo, E., Bianconi, R., Vautard, R., Appel, K. W., Moran, M. D., Hogrefe, C., … Galmarini, S. (2012). Model evaluation and ensemble modelling of surface-level ozone in Europe and North America in the context of AQMEII. Atmospheric Environment, 53, 60–74. https://doi.org/10.1016/j.atmosenv.2012.01.003

Sofen, E. D., Bowdalo, D., Evans, M. J., Apadula, F., Bonasoni, P., Cupeiro, M., Ellul, R., Galbally, I. E., Girgzdiene, R., Luppo, S., Mimouni, M., Nahas, A. C., Saliba, M., and Tørseth, K.: Gridded global surface ozone metrics for atmospheric chemistry model evaluation, Earth Syst. Sci. Data, 8, 41-59, https://doi.org/10.5194/essd-8-41-2016, 2016

---

## Author Response (AR2)

Dear Fiona,

Thank you very much for your comments and decision. We revised the manuscript to address your comments.

- Yes, the FAST-JX photolysis scheme is able to see aerosol loading. Aerosols are coupled to gas-phase chemistry via photolysis, heterogeneous chemistry, etc. In examining the manuscript, we note that the aerosol microphysics was not using TOMAS in this Nature Run, but a standard/default GEOS-Chem bulk aerosol scheme. I apologize for this error. We corrected it and clarified the photolysis - aerosol interaction in the manuscript as:
    o "The default GEOS-Chem bulk aerosol scheme is used to simulate major components for dust, sea salt, black carbon, organic carbon, sulfate, nitrate and ammonium aerosols (Park et al. 2004, Fairlie et al., 2007, Jaegle et al., 2011, Wang et al. 2014, Kim et al. 2015). The Fast-JX scheme with approximate randomized cloud overlap method and taking account for aerosol loading is used to calculate photolysis frequencies (Bian2002), as implemented by Mao et al. 2010."

- We added the Author contributions and Competing interests in the manuscript.
    o Author contributions. SP and DJJ provided project oversight and top-level design. CAK, MSL, BA, ADS, JEN, MAT, and ALT performed code development. KRT, SKG, and MJE provided additional data for model evaluation. LH, CAK, and TS performed evaluation analysis. LH, CAK, and DJJ wrote the manuscript. All authors contributed to manuscript editing and revisions.
    o Competing interests. The authors declare that they have no conflict of interest.

We appreciate you serving as the Editor of our manuscript and for recommending its publication.

Best regards,

Lu

[revised manuscript text omitted]